# Stochastic homeostasis in human airway epithelium is achieved by neutral competition of basal cell progenitors

Vitor H Teixeira[1], Parthiban Nadarajan[1], Trevor A Graham[2,3], Christodoulos P Pipinikas[1], James M Brown[1], Mary Falzon[4], Emma Nye[5], Richard Poulsom[2,6], David Lawrence[7], Nicholas A Wright[2,8], Stuart McDonald[2,8], Adam Giangreco[1], Benjamin D Simons[9,10,11]*, Sam M Janes[1]*

[1]Lungs for Living Research Centre, UCL Respiratory, University College London, London, United Kingdom; [2]Histopathology Laboratory, Cancer Research UK London Research Institute, London, United Kingdom; [3]Centre for Evolution and Cancer, UCSF Helen Diller Family Comprehensive Cancer Center, San Francisco, United States; [4]Department of Histopathology, University College Hospital London, London, United Kingdom; [5]Experimental Histopathology Laboratory, Cancer Research UK London Research Institute, London, United Kingdom; [6]Centre for Digestive Diseases, Blizard Institute, Barts and the London School of Medicine and Dentistry, Queen Mary, University of London, London, United Kingdom; [7]Department of Cardiothoracic Surgery, The Heart Hospital, London, United Kingdom; [8]Centre for Tumour Biology, Barts Cancer Institute, John Vane Science Centre, Barts and the London School of Medicine and Dentistry, Queen Mary, University of London, London, United Kingdom; [9]Cavendish Laboratory, Department of Physics, University of Cambridge, Cambridge, United Kingdom; [10]The Wellcome Trust/Cancer Research UK Gurdon Institute, University of Cambridge, Cambridge, United Kingdom; [11]Wellcome Trust–Medical Research Council Stem Cell Institute, University of Cambridge, United Kingdom

*For correspondence: bds10@hermes.cam.ac.uk (BDS); s.janes@ucl.ac.uk (SMJ)

**Abstract** Lineage tracing approaches have provided new insights into the cellular mechanisms that support tissue homeostasis in mice. However, the relevance of these discoveries to human epithelial homeostasis and its alterations in disease is unknown. By developing a novel quantitative approach for the analysis of somatic mitochondrial mutations that are accumulated over time, we demonstrate that the human upper airway epithelium is maintained by an equipotent basal progenitor cell population, in which the chance loss of cells due to lineage commitment is perfectly compensated by the duplication of neighbours, leading to "neutral drift" of the clone population. Further, we show that this process is accelerated in the airways of smokers, leading to intensified clonal consolidation and providing a background for tumorigenesis. This study provides a benchmark to show how somatic mutations provide quantitative information on homeostatic growth in human tissues, and a platform to explore factors leading to dysregulation and disease.

## Introduction

In adults, stem cells reside at the apex of proliferative hierarchies and, either directly, or through a sequence of terminal divisions, give rise to the specialist differentiated cells that provide the functional properties of tissue. To conserve their number, following division, one cell must on average remain in

**eLife digest** As air flows into our lungs, the lining of the nasal cavity, the throat and the rest of the respiratory tract prevents microbes, bacteria, dust and other small particles from entering the lungs. The lining of these airways is made up of many different types of cells, which must be continuously replaced as they become damaged. Experiments in mice have shown that cells called basal cells act as progenitor cells to keep the lining supplied with new cells. Progenitor cells are similar to stem cells: they divide to make, on average, one copy of themselves and one mature cell of another type (such as a secretory cell). This ensures that healthy supply of progenitor cells is maintained for the future. However, it is not clear whether this process takes place at the level of individual progenitor cells or as an average for a population of cells.

Teixeira et al. have now performed a study which shows that basal cells achieve this balance as a result of averaging. The study took advantage of the fact that cellular organelles called mitochondria have their own DNA, which gradually accumulates mutations over time. This makes it possible to identify groups of cells that are descended from a single progenitor cell because they will all contain the same mitochondrial mutation.

By studying lung tissue from seven individuals, Teixeira et al. were able to identify clusters of related cells and found that, as expected, the size of the clusters increased with age. And by applying a mathematical model across all the cells in the study, it was discovered that whenever one basal progenitor cell committed to a particular fate, another progenitor cell duplicated itself: however, this balancing process happened in a random manner across a large number of cells, and not at the level of individual progenitor cells. Interestingly, it was found random cell division happened among smokers too, but was accelerated. This leads to clusters of identical cells forming more quickly in smokers than in non-smokers. In addition to providing further insights into the origins of lung cancer, the statistical methods developed by Teixeira et al. could be used to analyse the behaviour of many other types of stem or progenitor cells.

the stem cell compartment while the other must commit to a differentiation pathway. This asymmetry may be enforced at the level of individual cells, or it may be achieved on a population basis, so that stem cell proliferation is perfectly compensated by the differentiation of others (*Clayton et al., 2007*; *Klein et al., 2010*; *Gomes et al., 2011*). In recent years there has been significant progress in defining common strategies of stem cell self-renewal using lineage tracing assays in mice (*Simons and Clevers, 2011*). By tracing the clonal evolution of marked cells following genetic pulse-labelling of transgenic animals, statistical methods have been used to discern the pattern of adult stem cell fate in several actively cycling mammalian tissues. In skin (*Clayton et al., 2007*; *Doupe et al., 2010*), oesophagus (*Doupe et al., 2012*), gut (*Lopez-Garcia et al., 2010*; *Snippert et al., 2010*) and testis (*Klein et al., 2010*), such methods have demonstrated that tissue maintenance involves the continuous stochastic loss and replacement of stem cells. However, although such approaches provide key insights into the mechanisms of stem cell maintenance in transgenic animal models, their relevance and application to human tissues is unknown.

To date, our understanding of human airway homeostasis is limited and inferred indirectly through in vitro studies of human cell culture and murine airway models (*Hong et al., 2004a*; *Schoch et al., 2004*; *Hajj et al., 2007*; *Rawlins et al., 2007*; *Giangreco et al., 2009*; *Rawlins et al., 2009*; *Rock et al., 2009*). In both humans and mice, airways are composed of basal, ciliated, secretory, and chemosensory cell populations. Collectively, these cells produce antimicrobial proteins, eliminate bacteria and toxic compounds via the mucociliary escalator, warm inspired air, and act as physical barriers to exogenous particulate matter (*Knight and Holgate, 2003*). Previous studies of human tissue have demonstrated that basal cells can exhibit multipotent growth and differentiation properties when grown in vitro under specific conditions, while in vivo murine studies demonstrate that basal cells of the major airways function as common multipotent progenitors (*Rock et al., 2009*, *2010*).

Using lineage tracing methods, involving chimeric tissue analysis (*Giangreco et al., 2009*) and genetic cell labelling (*Rawlins et al., 2009*), murine airways appear to be maintained in homeostatic state by abundant progenitor cells located throughout airways. However, the range and identity of the stem

cell compartment, and the frequency of stem cell loss and replacement, has not been quantified. Moreover, the relevance of these studies for the maintenance of human airways remains unexplored. In this study, we show that the continuous accumulation of somatic mutations that occur in the mitochondrial genome provides a clonal record from which the self-renewal properties of the human airway stem cell population can be inferred. As well as providing new insight into human airway maintenance, this study exemplifies a general methodological scheme that can serve as a template to study adult stem cell fate in other human tissues. Moreover, we show that the characterisation of normal tissue maintenance provides a quantitative platform from which we can study factors promoting the dysregulation of cells leading to the development of disease.

In order to directly assess the in vivo identity and potency of individual human airway epithelial progenitor cells, we make use of the predisposition of mtDNA to develop spontaneous mutations that affect expression of the cytochrome *c* oxidase (CCO) gene. CCO gene mutations occur spontaneously in all cells in a stochastic manner, do not significantly affect cellular function, and are unrelated to cellular toxicant exposure (*Elson et al., 2001*; *Taylor et al., 2001*; *Carew and Huang, 2002*; *Taylor et al., 2003*; *Taylor and Turnbull, 2005*; *Greaves et al., 2006*; *McDonald et al., 2008*; *Fellous et al., 2009*; *Gutierrez-Gonzalez et al., 2009*; *Lin et al., 2010*; *Gaisa et al., 2011a*; *Nicholson et al., 2011*). Thus, the division and accumulation of CCO-deficient cells leads to the formation of clonal patches of CCO-deficient cells within tissues, including the normal airway, and their examination provides a unique, histologically traceable record of airway progenitor cell fate. We use genetic sequencing to confirm the clonal origin of individual CCO patches and immunofluorescence to assess the cellular composition of these clones. Then, using statistical modelling of the frequency and size distribution of CCO-deficient clones visualised using whole mount labelling, we establish the cellular hierarchy and the in vivo pattern of airway homeostasis, making an explicit comparison between non-smokers and smokers.

From a detailed and quantitative analysis of the size and composition of CCO-deficient clones, we provide evidence that the maintenance of upper human airways relies upon multipotent progenitor cells that reside within the basal cell population. Further, we show that these cells maintain homeostasis through a process of population asymmetry in which their chance loss following commitment to differentiation is perfectly balanced by the duplication of others. This stochasticity leads to a natural process of age-associated airway clonal consolidation, which is notably accelerated in smokers, most likely due to increased rates of cellular turnover. As well as its intrinsic interest to human airway stem and progenitor cell biology, this study provides the benchmark to show how quantitative insights can be obtained from in vivo lineage tracing studies in human tissues, with obvious implications for studies of clonal progression in neoplasia.

## Results

### Phenotypic analysis of CCO-deficient patches is consistent with normal airway

To detect CCO-deficient cell patches of airway epithelial cells, we combined immunofluorescence labelling for CCO (*Figure 1A,B*, green), with the pan-mitochondrial protein porin (*Figure 1A,B*, red). Cells deficient in CCO, but marked by porin, indicate cell patches with CCO mitochondrial DNA mutation (*Nicholson et al., 2011*). Using lung whole-mount imaging of seven patients of varying age (*Table 1*), we identified and quantified CCO-deficient patches of cells within the third generation bronchi of human upper airways (*Figure 1A,B,C*). These patches were rare and randomly distributed within the airways (*Figure 1A*). Consistent with previous observations, no CCO-deficient individual cells, or patches of cells, were found in the 25 year old patient, despite examination of over one million cells, placing a constraint on the time taken for the chance clonal selection of a single mitochondrial mutation within an individual cell (*Greaves et al., 2006*). Within cells, there are thousands of mitochondria, each containing multiple copies of mtDNA. mtDNA mutations are random and increase with age (*Brierley et al., 1998*; *Michikawa et al., 1999*). Through chance expansion these mutations can be present in all copies of the mitochondrial genome (homoplasmy) or a proportion thereof (heteroplasmy). For the mutated mitochondrial CCO genotype to result in a loss of CCO expression, homoplasmy or high levels of heteroplasmy must be present (*Sciacco et al., 1994*).

Normal lung airway contains two regions. The upper airway epithelium contains basal, ciliated, and goblet cells (*Figure 1D*), while the lower conducting airways include Clara cell secretory protein positive

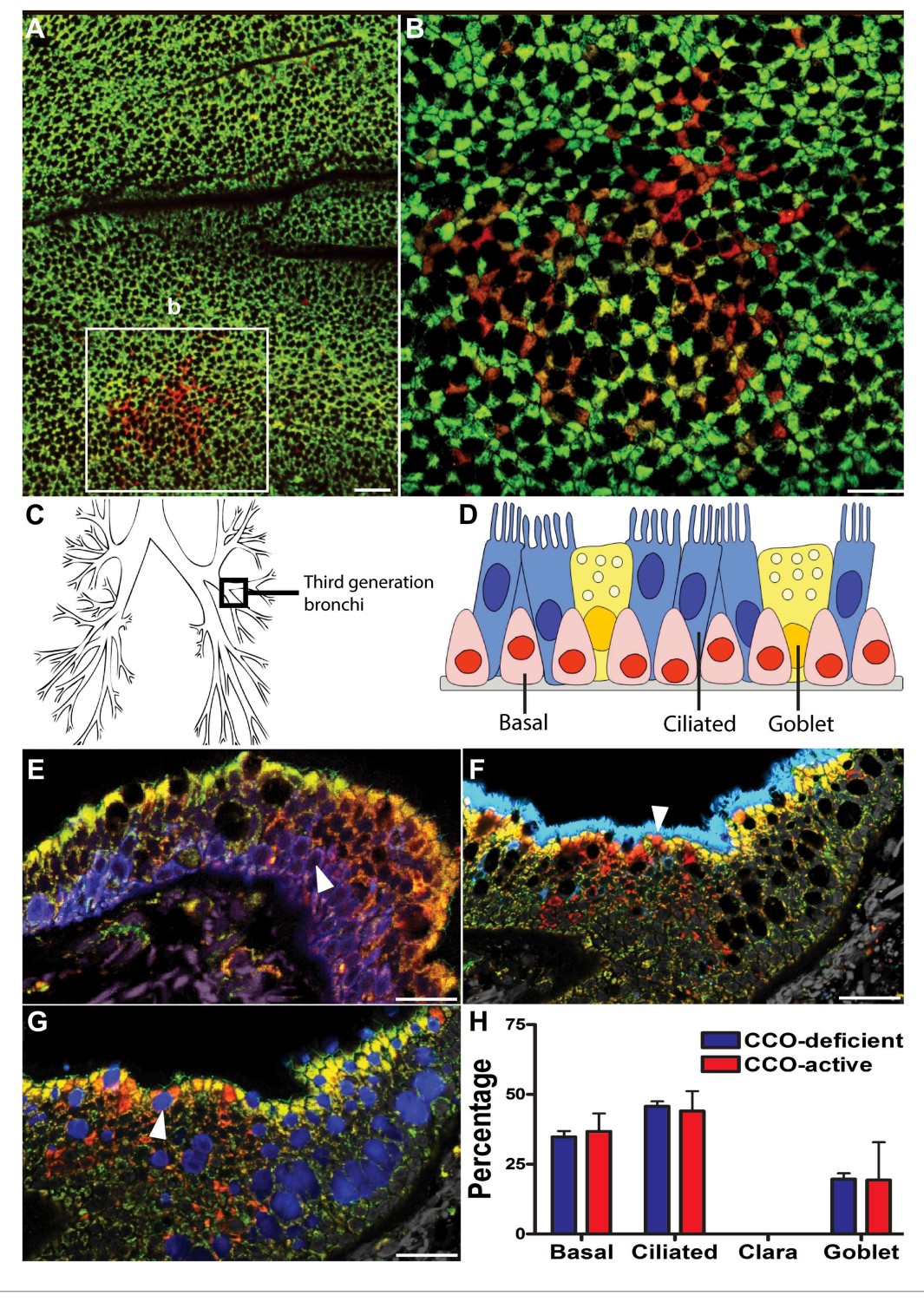

**Figure 1**. CCO-deficient human epithelial patches in the upper airway demonstrate multipotent differentiation. (**A** and **B**) Third generation bronchi stained for CCO (green) and counterstained with mitochondrial marker porin (red) show CCO-deficient patches. The black spaces are Goblet cells with vesicles containing mucus. (**C**) Schematic showing the location of the third generation human bronchi. (**D**) Schematic showing lung epithelial cell types of the upper airways. (**E**) CCO-deficient clonal patches contain keratin 5 positive basal cells (blue; arrowhead), (**F**) acetylated tubulin positive ciliated cells (blue; arrowhead) and (**G**) Mucin 5AC positive goblet cells (blue: *Figure 1. Continued on next page*

*Figure 1. Continued*

arrowhead). (**H**) There was no difference in lineage-specific differentiation between the CCO-deficient patches and the CCO-active lung epithelial cells. Cell percentages were calculated after counting all cells from 11 CCO-deficient patches and 11 CCO-active patches (basal cells–34.7 ± 2.1 vs 36.7 ± 6.5; ciliated cells–45.7 ± 1.7 vs 44 ± 7.2, Goblet cells–19.6 ± 2.1 vs 19.3 ± 13.5 [mean ± SD]). Scale bars—50 µm.

The following figure supplements are available for figure 1:

**Figure supplement 1**. CCO-deficient human epithelial patches in the lower airway demonstrate multipotent differentiation.

**Figure supplement 2**. c-kit cells stained (red) did not stain with epithelial markers (green) and were not present in most patches.

cells (Clara cells). To confirm a normal epithelial make up of the epithelial CCO-deficient patches we performed immunostaining of the two regions. These studies revealed that large CCO-deficient patches in upper human airways contain the expected bronchial epithelial cell types including keratin 5 positive basal cells, acetylated tubulin positive ciliated cells, and Mucin 5AC positive goblet cells (*Figure 1E–G*). Based on the percentage of constituent cell types, we found that there was no significant difference in cell type composition between CCO-deficient clonal patches and the neighbouring CCO-active lung epithelial cells (*Figure 1H*). On examination of the lower conducting airways, the frequency of Clara cells in CCO-active lung epithelial cells matched with that of the neighbouring tissue (*Figure 1—figure supplement 1*). Of note, we observed rare c-kit positive cells, recently proposed as a putative marker for pulmonary stem cells in the airway. However, these c-kit positive cells were stained uniformly with CD45 (leucocyte common antigen), but not epithelial markers, and they were absent in most patches (*Figure 1—figure supplement 2*).

## Cells within a single CCO-deficient patch are clonal, while each patch is genetically distinct from each other

Cells within individual CCO-deficient patches were demonstrated as clonally derived by single cell laser microdissection and subsequent mitochondrial genome sequencing (*Figure 2A–F*). Mitochondrial sequencing of single cells requires frozen tissue, and patches were identified on frozen tissue sections using dual colour enzyme histochemistry, simultaneously detecting enzyme activity of the mtDNA-encoded CCO and nuclear DNA-encoded succinate dehydrogenase (SDH). CCO-deficient cells, or patches of cells (stained blue), were seen to be surrounded by CCO-expressing adjacent lung epithelial cells (stained brown), and individual cells were laser dissected (*Figure 2A,D*).

Within the patch shown in *Figure 2A*, a 9850 T>C homoplasmic mutation was found in all CCO-deficient lung epithelial cells (*Figure 2B*) while all CCO-normal cells maintained their wild-type genotype (*Figure 2C*). This 9850 T>C mtDNA mutation is present in the mitochondrial cytochrome *c* oxidase III gene, and results in a Leu215Pro amino acid substitution in cytochrome *c* oxidase protein predicting the identified CCO deficiency. The second patch, shown in *Figure 2D–F*, also demonstrated that CCO-deficient lung epithelial cells shared the same mtDNA mutation not present in adjacent CCO-normal

**Table 1.** Patient characteristics

| Patient | Age (Surgery) | Sex | Smoking (pack years) |
|---|---|---|---|
| 1 | 39 | M | 30 |
| 2 | 55 | F | 40 |
| 3 | 79 | M | 40 |
| 4 | 25 | F | Non smoker |
| 5 | 47 | M | Non smoker |
| 6 | 57 | F | Non smoker |
| 7 | 65 | M | Non smoker |

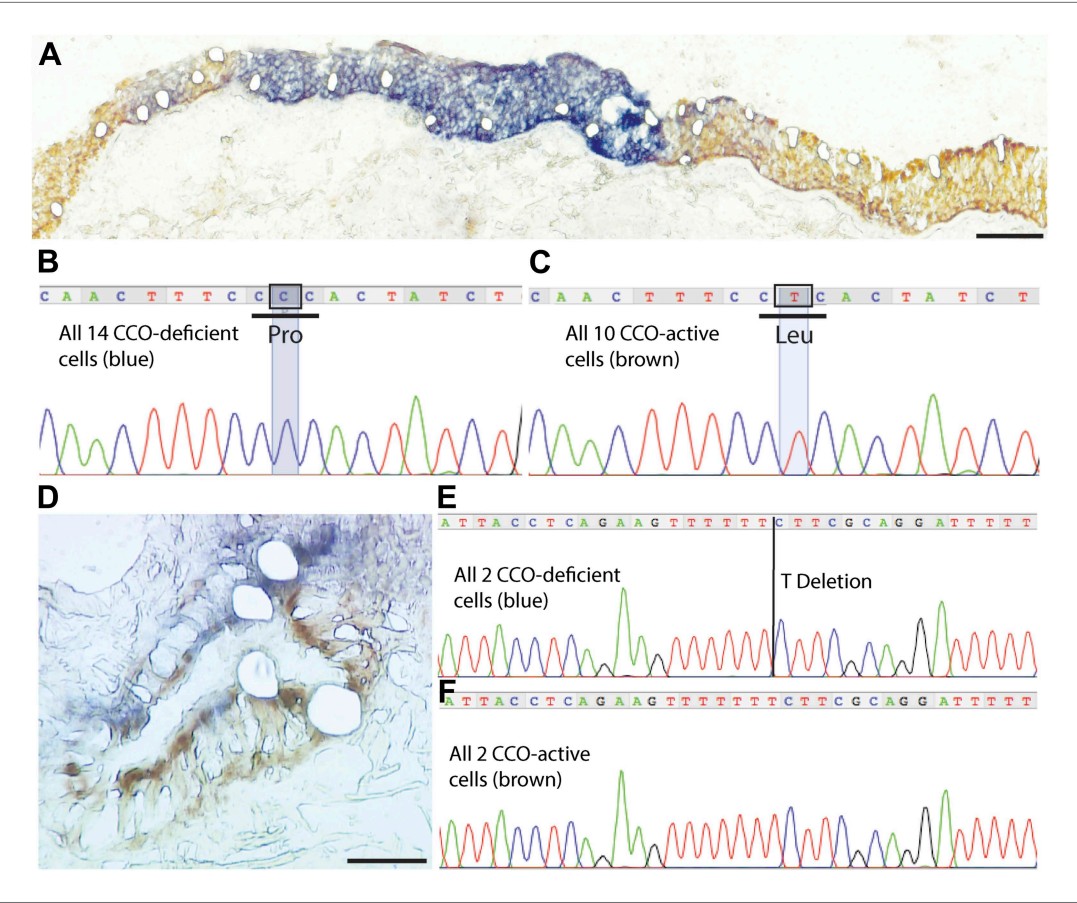

**Figure 2**. CCO-deficient patches are clonal with distinct mutations from neighbouring patches. (**B** and **D**) Histochemistry demonstrates CCO-deficient patch (dark blue) surrounded by CCO-active adjacent lung epithelial cells (brown) with individual cells laser micro-dissected. (**B** and **E**) Homoplasmic mutation shared by all CCO-deficient (blue) lung epithelial cells both within the main patch and blue cells scattered around the edges of the patch. (**C** and **F**) CCO-active cells show wild-type genotype. Scale bars—100 μm.

cells. In total, seven CCO-deficient patches of varying size from three patients were genetically analysed and confirmed that each patch was clonally distinct, as evidenced by patch-unique mtDNA mutations (*Table 2*). This demonstrates clonal expansion of proximate epithelial cells within morphologically normal airways.

## Patch analysis demonstrates basal cells contain the multipotent airway progenitor

There was no significant difference in the proliferation index (as measured by Ki67) between CCO-positive epithelium and CCO-negative patches (*Figure 3A,B*). Significantly, on examining the cell types within CCO-deficient patches, only ciliated and basal cells were always present in CCO-deficient patches with more than five cells, consistent with murine data identifying basal cells as the most likely host of multipotent progenitors of the upper airway (*Figure 3C,D*). While the majority of clones with five cells or less contain one or more basal cells (*Figure 3E*), a few small clones lack basal cells altogether suggesting basal progenitors are able to undergo terminal division leading to clone loss (*Figure 3— figure supplement 1A–B*). Indeed, this behaviour is corroborated by measurements of the clone density. In total we identified 844 CCO-deficient patches among the 12.7 million cells studied in the seven patients. However, despite the ongoing marking of cells through sporadic mutation, the number of surviving clones does not increase substantially between 39 to 79 year old patients (*Figure 4A*). As well as providing further evidence of clone loss through differentiation, this result suggests that the frequency of loss and replacement may be high.

**Table 2.** Analysis of mtDNA mutations of seven CCO-deficient patches, from three patients

| Patient | Patch | CCO-deficient cells mutation | Gene |
|---------|-------|------------------------------|------|
| 1 | 1 | 9850 T>C | MT-CO3 |
| | 2 | 6708 G>A | MT-CO1 |
| | 3 | 6838 T>C | MT-CO1 |
| | 4 | 6690 G>A | MT-CO1 |
| | 5 | 6692 A deletion | MT-CO1 |
| 2 | 6 | 9478 T deletion | MT-CO3 |
| 3 | 7 | 6087 G>A | MT-CO1 |

Multiple single cells laser captured from each patch confirmed the same sporadic mutation and hence patch clonality.

Although these findings are consistent with the existence of a multipotential self-renewing stem cell within the basal cell population, they leave open the question of stem cell identity, potency, and pattern of fate. To classify the behaviour of the airway progenitor cell population, we turned to a more detailed quantitative analysis of the size distribution of CCO-deficient clones in the upper airways by analysing large areas of whole-mount airways.

## Clonal fate provides evidence that the maintenance of upper human airways involves population asymmetric self-renewal

To develop a more quantitative analysis of the clonal fate data, we recorded in detail the total sizes of CCO-deficient clones for all patients including smokers and non-smokers. With clonal densities of around 100/cm$^2$ (*Figure 4A*), we estimate a typical separation between clones to be around 100 cell diameters. With typical clone sizes of around 100 cells or less, this suggests that errors due to clone merger events are likely to play only a minor role. The results, shown in *Figure 4B,C*, are depicted through the cumulative clone size distribution, $C_n(t)$, representing the frequency of clones larger than size $n$ cells. For example, referring to the 39 year old smoker, some 25% of clones have a total size of more than 10 cells, etc. From these studies, we find a broad distribution of clone sizes with clones as large as 30–40 cells coexisting with clones with only one or two cells. While the clones show a small drift to larger sizes with increasing age, the distributions remain broad.

Although the characterisation of CCO-deficient clones by size provides indirect access to the fate behaviour of marked cells and their progeny, the interpretation of these data sets is not straightforward. First, as a result of the sporadic nature of cell labelling, the clonal history of 'young' clones is not readily disentangled from that of clones induced in the distant past. For example, small clones may derive from recently induced cells or they may be associated with chance expansion and contraction through differentiation and cell loss in older early marked cells. Furthermore, the interpretation of the clonal fate data, and the association of clone size with a measure of the number of constituent progenitor cells in the clone, may be further complicated by the existence of a transit-amplifying cell hierarchy. Therefore, to critically assess the clonal data for signatures of equipotency and progenitor cell fate, it is useful to develop a simple and robust biophysical modelling scheme, the validity of which can be checked self-consistently through characteristics in the clone fate data.

To develop our model, let us suppose that maintenance of lung epithelium involves the steady turnover of a single, multipotent, functionally equivalent (i.e., equipotent), tissue-maintaining population, a necessary condition of long-term tissue homeostasis. Following division of these proliferative cells, the daughters can adopt one of three possible fate outcomes—two progenitor cells, two cells that have committed to a differentiation pathway (either directly or through a series of terminal divisions), or one progenitor and one differentiating cell (*Figure 5A*). At this stage, we do not discriminate between different differentiating cell types and, in doing so, presume that the balance between proliferation and differentiation is not correlated with particular fate choice. To ensure long-term homeostasis, symmetric divisions leading to progenitor cell duplication must be perfectly balanced by those leading to differentiation and loss. In this paradigm, clones are predicted to follow a process of 'neutral drift' in which the chance expansion of some clones through proliferation is compensated by the contraction and extinction of others through differentiation and subsequent loss (*Figure 5B*).

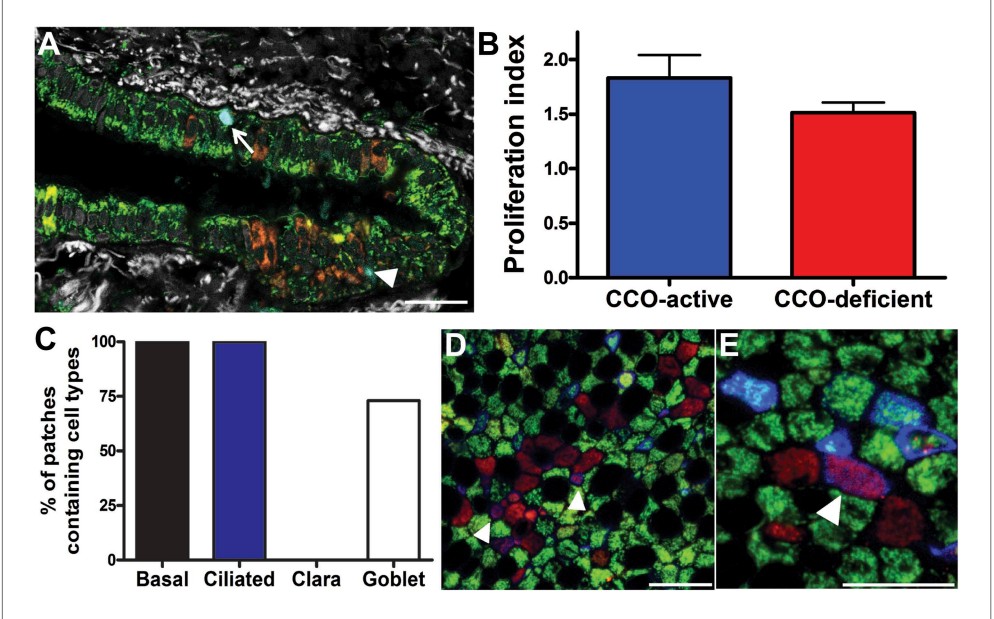

**Figure 3**. CCO-deficient upper airway patches are representative of the upper airway. (**A**) Ki67 immunofluorescence showing a positive CCO-deficient cell (arrowhead) and a positive CCO-active cell (arrow). (**B**) There was no difference in the proliferation index between CCO-deficient clonal patches and CCO-active lung epithelia. CCO-active patches (n = 11) vs CCO-deficient patches (n = 11) (mean ± SD, 1.8 ± 0.2 vs 1.5 ± 0.1). (**C**) Cell type examination of patches greater than five cells showed that only ciliated and basal cells were present in all CCO-deficient patches. (**D**) Shows a clone greater than five cells with basal cells—arrowhead points to violet basal cells due to an expression of Porin (red) and Krt5 (blue) within a CCO-deficient patch. (**E**) Shows a small clone with a basal cell—arrowhead points to violet basal cells due to an expression of Porin (red) and Krt5 (blue) within a CCO-deficient patch. (CCO, green; Porin, red; Keratin 5, blue). The black spaces in **D** and **E** are Goblet cells with vesicles containing mucus. Scale bars—50 μm.

The following figure supplements are available for figure 3:

**Figure supplement 1**. Shows rare small clones with no basal cells.

Following this pattern of stochastic fate choice, previous studies have shown that the size distribution of clones derived from a single progenitor cell converge onto a scaling form in which the chance of finding a surviving clone with $n>0$ progenitor cells after a time $t$ post-labelling is given by, $P_n^{surv\cdot}(t) = (1/N(t))exp[-n/N(t)]$, with the average size, $N(t) = (1+r\lambda t)/\rho$, growing linearly with time (**Clayton et al., 2007**). Here $\lambda$ denotes the progenitor cell division rate, $r$ specifies the balance between symmetric and asymmetric fate outcome (**Figure 5A**), and $\rho$ defines the total number of progenitor cells as a fraction of the total cell population. This growth in the average size of surviving clones perfectly compensates for clones that are lost through commitment to differentiation so that the average number of labelled progenitor cells remains fixed at one per induced clone. Note that, since the proliferative capacity of any transit-amplifying cell progeny is strictly limited, its effect can influence the value of $\rho$, but does not effect the long term scaling form of the size distribution.

In the present case, since mitochondrial DNA mutations led to a steady accumulation of marked cells over time, the effective size distribution of surviving clones involves the sum over all of these histories (**Figure 5C**), and is predicted to take the form (**Klein et al., 2010**),

$$P_n^{surv\cdot}(t) = \frac{1}{ln[\rho N(t)]} \frac{exp[-n/N(t)]}{n}. \qquad (1)$$

Here $N(t)$ denotes the average size reached by surviving clones induced in the first round of visible mutations (i.e., after a time, t).

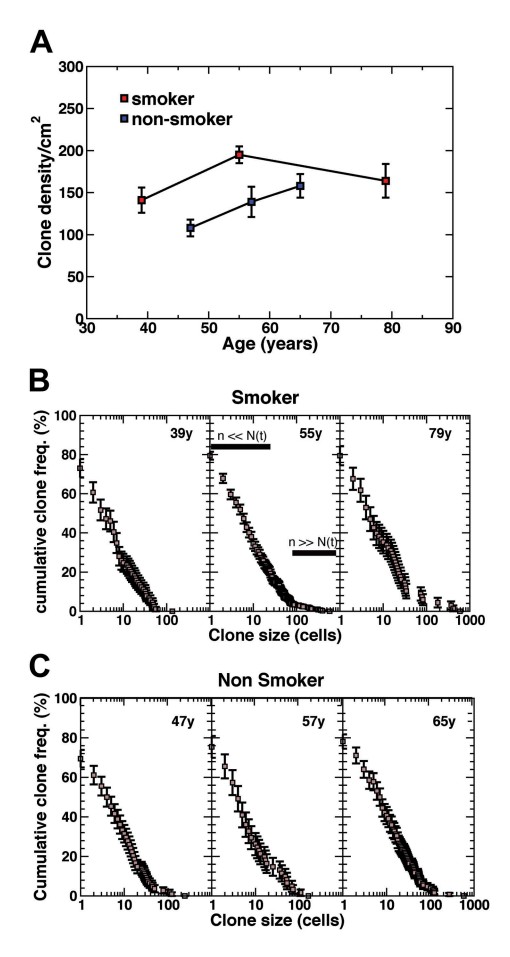

**Figure 4**. Density of CCO-deficient clones and their size distribution. (**A**) Measured clone density (number per unit area) of CCO-deficient clones measured in three smokers and three non-smokers of different ages. Note that, with increasing age of patients, the clone density changes little, while the overall density in the non-smoker is significantly smaller than smokers of a similar age. (**B** and **C**) Cumulative clone size distribution of CCO-deficient clones, $C_n(t)$, for the three smokers (**B**) and three non-smokers (**C**) showing the probability that a clone has a size larger than $n$ cells in patients of age $t$ = 39 years, 55 years, and 79 years for the smokers and 47 years, 57 years and 65 years for the non-smokers. (Errors denote SEM).

The following figure supplements are available for figure 4:

**Figure supplement 1**. Raw clone size data.

Although the predicted size dependence, **Equation 1**, can be immediately compared with the observed clonal fate data (**Figure 4B,C**), it is important to consider which parts of the dataset are valuable (**Klein et al., 2010**). For $n \ll N(t)$ (approximately **Figure 4B**, 55y), the form of the clone size distribution will be dominated by clones created in the 'recent past'. For these clones, the sum over possible histories (i.e., clone birthdates) masks the divergence of individual clone sizes due to stochastic fate choice, and leads to the featureless $1/n$ dependence that dominates the distribution, **Equation 1**, for small $n$. However, for $n \geq N(t)$ (approximately **Figure 4B**, 55y), the clone size distribution becomes sensitive to the 'front' of clones that were marked at the earliest time point. In the present context, this translates to the age at which the CCO-deficient clones first become visible. In this limit, the clone size distribution, **Equation 1**, is dominated by the exponential dependence.

From a fit of the model prediction, **Equation 1**, to the raw experimental data (**Figure 4B,C**, **Figure 4—figure supplement 1**), we find excellent agreement (**Figure 6A,B**, **Figure 6—figure supplement 1**) for all six patients (smokers and non-smokers) with CCO-deficient clones, with the inferred values of average clone size $N(t)$, shown in **Figure 6C**. (Here, to amplify the tail of the clone size distribution, we have studied the behaviour of a 'derivative' of the cumulative distribution known as the first incomplete moment. For further details, see 'mathematical analysis') From the fit of the experimental data with the predicted form of the distribution, we conclude that the multipotent progenitor cells that line the lung epithelium of the upper airways form a single, equipotent population that maintain tissue through a process of population asymmetry. In the course of turnover, the chance loss of progenitors through differentiation is perfectly compensated by duplication of neighbouring progenitors leading to a neutral drift in the size of surviving clones and a continual depletion in clonal diversity of tissue (**Figure 5A**).

## The frequency of progenitor cell loss and replacement is enhanced in smokers

As well as providing a signature of population asymmetry, from the analysis of the average clone size, $N(t)$, and clone density, we can infer the frequency of progenitor cell loss and replacement. In particular, for both smokers and non-smokers, the values of $N(t)$, inferred from the quantitative analysis of the clone fate data follow approximately the predicted linear time-dependence (**Figure 6C**), with a rate constant of $r\lambda/\rho$ = 2.7 ± 0.5/year for smokers and $r\lambda/\rho$ = 1.5 ± 0.8 for non-smokers. Moreover, from the extrapolation of $N(t)$ to zero, we can deduce that CCO-deficient mutations become visible in patients at approximately 20 ± 10 years of age,

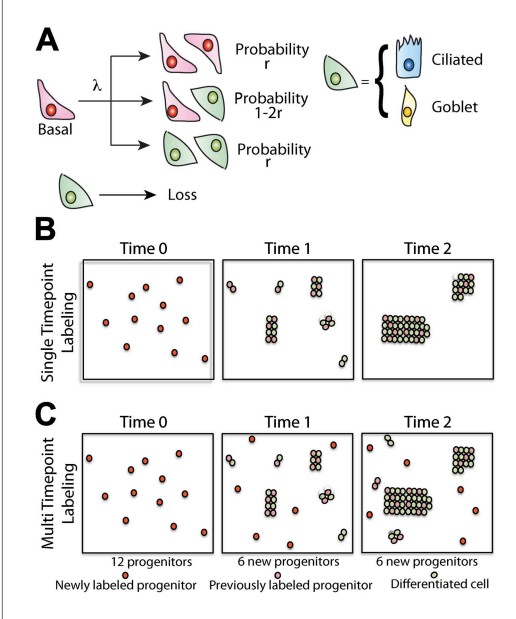

**Figure 5**. Maintenance of lung epithelium involves neutral competition. (**A**) Schematic showing the model hypothesis used to interpret and analyse the clonal fate data. According to this model, maintenance of the human airway epithelium involves the balanced stochastic fate of tissue-maintaining cells in which cell division results in all three fate outcomes: symmetric duplication, asymmetric division, or symmetric differentiation. λ denotes the corresponding cell division rate, and r controls the fraction of divisions that lead to symmetric fate outcome. Note that differentiation can lead to any one of the differentiated cell types, with probabilities commensurate with the tissue composition. Here, for simplicity, we associate tissue-maintaining cells with the basal progenitor population. However, if tissue-maintaining cells constitute only a subpopulation of these basal progenitors, we would obtain the same long-term clone fate behaviour, *Equation 1*, while the overall fraction of tissue-maintaining cells, ρ, would have to be adjusted accordingly. (**B**) Schematic depicting the pattern of clonal evolution following pulse-labelling of tissue. As tissue turns over, chance clonal loss is perfectly compensated by the expansion of other clones so that the overall number of labelled progenitor cells remains approximately constant—a process reminiscent of 'neutral drift'. (**C**) Through the spontaneous acquisition of somatic mtDNA mutation single cells become clonally marked throughout adulthood. As result, the clones distribution at any given time point represents the amalgamation of clones of different ages from those induced at the earliest times when CCO-deficient cells first become visible, to those marked within the recent past.

consistent with the absence of CCO-deficient patches in the 25 year old patient.

If we suppose that the basal cell population comprises the entire population of tissue-maintaining progenitor cells, from their proportion of the total lung epithelial cell population (*Figure 1H*) we can estimate $\rho = 1/3$, from which it follows that the progenitor cell loss/replacement rate is given by $2r\lambda = 1.8 \pm 0.4$/year. Intriguingly, this compares to the inferred value for non-smokers of $2r\lambda = 1 \pm 0.5$/year, a factor of around two smaller. Further support for this conclusion is found from the observation of single cell clones, which are more prevalent in non-smokers than smokers (*Figure 4—figure supplement 1*), consistent with a higher rate of turnover in the later. Although this methodology provides the means to estimate the progenitor cell loss/replacement rate, the total cell division rate is not accessible. Cell divisions leading to asymmetric fate have no impact on clone size. As a result, we cannot say whether the acceleration of loss/replacement in smokers is simply a reflection of enhanced proliferation, or represents a tilt in the balance between symmetric and asymmetric cell division.

Alongside the clone size distribution, the neutral drift model also predicts the accumulation rate of surviving clones. In homeostasis, for a constant mutation rate, $R$, the surviving clone density (clones per unit area of tissue) is predicted to rise only slowly (logarithmically) over time varying according to the relation, $\sigma = (\rho R/r\lambda)\ln\left[N(t)\right]$ ('Mathematical analysis'). Such a slow growth is a manifestation of the neutrality of clonal evolution leading to the chance loss of the majority of induced progenitors. This dependence of clone density on the parameters of cell fate can be used to further challenge the model. From the small increase in clone density in smokers from 39 to 79 years (*Figure 4A*), as expected from the predicted logarithmic growth dependence, using the inferred values of $r\lambda/\rho$ and $N(t)$, we find a mutation rate of $R = 100 \pm 10$/year/cm². With a cell density of around $10^6$/cm², this translates to the acquisition of CCO deficiency at a rate of around $10^{-4}$/cell/year. By contrast, applied to the data for non-smokers from age 47 to 65 years, we find a mutation rate with a lower figure of $R = 60 \pm 10$/year/cm².

## Discussion

In summary, these results show that maintenance of human lung epithelium involves the ongoing stochastic loss and replacement of a single, functionally equivalent, multipotent, progenitor cell population. Although the clonal fate data does not unambiguously disclose the identity of this population, the findings are consistent with tissue-maintaining

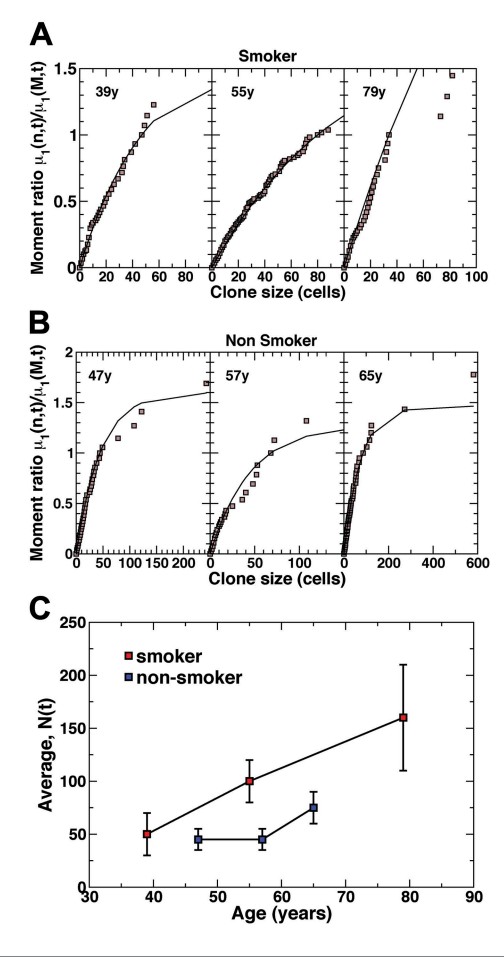

**Figure 6**. Quantitative analysis of the clonal fate data. Comparison and fit of the first incomplete moment, a derivative of the cumulative clone size distribution (*Figure 4B,C*), to the model prediction, *Equation 1* (for details, see 'Mathematical analysis') for (**A**) smokers and (**B**) non-smokers. Points show data and lines show the model prediction. From these comparisons, we obtain the one fitting parameter, *N(t)*, defining the average size of clones created at the earliest time point, as described in the main text. The resulting values of *N(t)* from these fits are shown in (**C**) for the three smokers and three non-smokers. Error bars depict the approximate range of *N(t)* values that provide satisfactory fits to the measured clonal fate data. Note that these inferred values are consistent with the predicted linear dependence of *N(t)* on age *t* providing further corroboration of the model dynamics.

The following figure supplements are available for figure 6:

**Figure supplement 1**. First incomplete moment distribution, derived from the cumulative clone size distribution.

cells belonging to the pool of basal progenitors. Moreover, this study provides strong evidence that the rate of progenitor cell loss and replacement is increased in smokers, providing a background on which the process of field cancerisation can operate.

In this study we exploited the natural occurrence of mitochondrial DNA mutations as markers of clonal expansion. Since only stem cells have long enough life spans to accumulate mtDNA mutations to detectable levels of homoplasmy, a clonal population marked by a particular mtDNA mutation likely represents the progeny of a single mutated stem cell (*Elson et al., 2001*; *Greaves et al., 2006*; *Fellous et al., 2009*). Using this lineage tracing methodology, stem cell niches have been located in many human tissues (*Elson et al., 2001*; *McDonald et al., 2008*; *Fellous et al., 2009*; *Gutierrez-Gonzalez et al., 2009*; *Lin et al., 2010*; *Gaisa et al., 2011b*). Our use, for the first time, of whole-mount human lung imaging allowed us to use quantitative analysis of these clones to demonstrate both that basal cells are the likely multipotent progenitor cells in the human upper airways and that airway homeostasis is maintained in a stochastic manner. The demonstration that a few small clones lack basal cells is consistent with terminal differentiation of a stem cell within a said clone, which would eventually lead to its loss.

Curiously, using elegant lineage tracing studies following injury, the presence of both unipotent and multipotent basal cells airway progenitors have been proposed in mice (*Hong et al., 2004b*). By contrast, the absence of large clones (>5 cells) with only basal cells suggests that, at least under normal homeostatic conditions, the basal population of the upper airways does not include a long-lived unipotent progenitor. However, further studies of the murine system, using 3D whole mount imaging of homeostatic tissue is required to establish whether these differences are genuine.

Lastly, we have systematically examined the upper large airways and do not discard a role for other cell types such as Clara cells in the maintenance of homeostasis of smaller human airways. In our whole mount imaging of large bronchi Clara cells are virtually absent, as found by others (*Boers et al., 1998*), but in distal conducting airway epithelium of normal human lungs Clara cells make up between 11 and 22% of the epithelial population and 9% of the overall proliferative compartment (*Boers et al., 1999*). In analysing the data, we have placed emphasis on a model in which progenitor cell fate choice follows from cell-autonomous (intrinsic) regulation, that is it is insensitive to environmental factors. However,

previous studies have shown that, in the two-dimensional geometry pertinent to the lung epithelium, a process in which progenitor cell loss was compensated by duplication of neighbouring progenitors through a coordinated process would lead to the same long-term exponential clone size distribution considered here, while the average clone size would acquire a small additional logarithmic dependence on time, which would not be resolved by the current data (*Klein et al., 2010*). As such, the question of the regulatory control underlying stochastic fate specification and balance is left open. In addition, these findings do not rule out the potential for a stem/progenitor cell hierarchy, with progenitors supported by a second quiescent stem cell population, as recently resolved in normal interfollicular epidermis (*Mascre et al., 2012*).

Finally, as well as its implications for the maintenance of the human airway epithelium, this study provides a benchmark to show how clones derived from the acquisition of somatic mutations can be used quantitatively to explore the pattern of homeostatic growth in other human tissues, providing a platform to investigate factors leading to dysregulation and disease.

## Material and methods

### Patients and ethical approvals

Human lung lobes were obtained from patients undergoing lung resection. Airways were taken at least 5 cm away from any tumour. The airways were histologically analysed and found normal. Specimens were either fixed in 4% paraformaldehyde and embedded in paraffin or snap-frozen in liquid nitrogen cooled isopentane. Ethical approval was sought and obtained from University College London Hospital Research Ethics Committee (REC reference 06/Q0505/12). This study was carried out in accordance with the Declaration of Helsinki (2000) of the World Medical Association.

### Histochemistry

Frozen lung samples were mounted in OCT compound for sectioning. Same section sequential histochemical staining for cytochrome $c$ oxidase (CCO), a component of complex IV of the respiratory chain enzyme, and succinate dehydrogenase histochemistry, a component of complex II of the respiratory chain (the presence of which was used to highlight the absence of CCO activity) was carried out on 10 µm transverse sections as follows. Sections were first incubated in CCO incubation medium (100 µM cytochrome $c$/4 mM diaminobenzidine tetrahydrochloride/20 µg/ml catalase in 0.2 M phosphate buffer, pH 7.0) for 50 min at 37°C. They were then washed in PBS, pH 7.4, for 3 × 5 min and incubated in SDH incubation medium (130 mM sodium succinate/200 µM phenazine methosulphate/1 mM sodium azide/1.5 mM nitroblue tetrazolium in 0.2 M phosphate buffer, pH 7.0) for 45 min at 37°C. Sections were washed in PBS for 3 × 5 min, dehydrated in a graded ethanol series (70%, 95%, and 100%) and left to air dry for 1 hr.

### Isolation of total DNA from individual cells

Single cells from CCO-deficient and CCO-positive airways were cut into sterile 0.5-ml PCR tubes using the Leica (Deerfield, IL, USA) Laser Microdissection (AS-LMD) System. Cell digestion and DNA extraction were performed by overnight incubation in a DNA extraction kit (PicoPure, Arcturus; Molecular Devices, Sunnyvale, CA, USA) at 65°C and then 95°C for 10 min to denature the proteinase K.

### mtDNA sequencing of individual lung epithelial cells

The extracted DNA was used to sequence the entire mitochondrial genome from microdissected areas. A two-round amplification method was followed, whereby the first round consisted of amplifying nine fragments spanning the entire genome, and the second round consisted of 36 M13-tailed primer pairs to amplify overlapping segments of the first-round products. Sequencing was performed using the BigDye terminator cycle sequencing method on an ABI Prism Genetic Analyzer (Applied Biosystems, Foster City, CA) and compared with the revised Cambridge reference sequence using sequence alignment software of the European Molecular Biology Open Software Suite (EMBOSS, http://www.ebi.ac.uk/emboss/align/).

### Immunofluorescence

4 µm formalin fixed paraffin lung sections were cut. Human OxPhos Complex IV Subunit Monoclonal antibody (CCO) (Invitrogen, Carlsbad, CA), Porin (Abcam, Cambridge, UK), Clara Cell Secretory Protein (gift from Barry Strip's lab—Goat anti—CCSP, Goat number 899), Keratin 5 (Abcam), acetylated tubulin (Sigma-Aldrich, St. Louis, MO), Ki67 (Dako, Glostrup, Denmark), Mucin 5AC (Sigma), c-kit (Dako) and cd45

(Dako) antibodies were used for immunostaining following EDTA (pH 9.0) antigen retrieval. Secondary antibodies included appropriate species-specific Alexa488, Alexa555, and Alexa633 dyes (Invitrogen). Isotype-matched and without primary controls revealed no nonspecific staining. Images were obtained using a Leica TCS Tandem confocal at 10x, 20x and 40x objective magnification (Leica Microsystems, Milton Keynes, UK). Small patches were assessed for cell type by contiguous sectioning.

## Whole-mount dissection and immunostaining

The samples mathematically analysed using the whole mount technique were from the third generation bronchi. Lobectomy specimens were used that had two clear features. First the surgical cut was across the second generation bronchus (meaning the upper on the right and left; or lower lobe on the left; or lower or middle lobe on the right). The resection margin was retained by pathology, and the next/third generation bronchi dissected out by the research team. Second any resected tumour had to be in the distal lung parenchyma to avoiding close proximity to the whole mount area. Importantly the distal end of the samples were blocked and shown to be normal to regular histology assessment. Human lung lobes were dissected open to expose the airways to antibodies staining. Whole-mount airways were fixed with 4% vol/vol paraformaldehyde, washed in 0.1 Triton X-100 in PBS, subjected to antigen retrieval by boiling in 10 mM citrate buffer for 40 min, and incubated in blocking buffer (3 hr; 10% FBS, 0.05% fish skin solution in PBS). Airways were then incubated overnight with antibodies against OxPhos Complex IV Subunit Monoclonal antibody (CCO) and Porin and were then washed in PBS (0.05% Tween 20) and incubated with Alexafluor-488 or -555—conjugated anti-mouse or anti-rabbit secondary reagents (Invitrogen) for 6 hr and then washed. Nuclei were stained with DAPI. Images were acquired using an Olympus epi-fluorescence microscope. A total number of 12.7 million cells included in seven different airways were analysed using the 100X objective of the Olympus epi-fluorescence microscope. Among them 20,002 CCO-deficient cells included in 844 CCO-deficient patches were identified.

## Mathematical analysis

To analyse the clonal fate data, we have made use of a simple biophysical modelling scheme. In keeping the focus of the main text on the principle results, important aspects of the mathematical analysis have been sacrificed. Here, in this section, we develop the modelling scheme more fully, stating clearly the assumptions on which it relies, and the experimental consistency checks that we can make to validate the approach. The method is one that is inspired by an earlier study on the time-evolution of p53 mutant clones in epidermis following exposure to UV-B radiation (*Klein et al., 2010*). This section is divided into three sub-sections. In the first, we consider the potential pattern of clonal growth in a homeostatic tissue following 'pulse-labelling' of cells. In the following section, we discuss how these results must be revised to accommodate on-going induction due to the continuous accumulation of sporadic mtDNA mutations. The application of these results is discussed in the main text and figures.

## Strategies of self-renewal

To address the lineage tracing data, we need a platform to interpret the clonal evolution. In the following, we will assume that, in adult, the lining of the human airway epithelium undergoes a slow and constant rate of turnover. Later, we will present experimental evidence in support of this assumption. Further, let us assume that the tissue-maintaining cell population is functionally equivalent (i.e., equipotent), a necessary condition of long-term homeostatic turnover (*Klein and Simons, 2011*), and able choose any of the three possible fate outcomes following division -symmetric duplication, asymmetrical cell division in which one of the progeny commits to a differentiation pathway, or terminal division in which both cells undergo commitment (*Figure 5A*).

Formally, we can denote this behavior by the process,

$$S \xrightarrow{\lambda} \begin{cases} S+S & \text{Pr}.\, r \\ S+D & \text{Pr}.\, 1-2r, \quad D \xrightarrow{\Gamma} \varnothing, \\ D+D & \text{Pr}.\, r \end{cases} \qquad (2)$$

where λ denotes the division rate of proliferative cells, S, $r$ controls the balance between symmetric and asymmetric cell fates, and Γ represents the rate at which differentiated cells, D, are lost (denoted

as Ø). Here we do not discriminate the different differentiating cell types, D Rather, we assume that, once a cell has committed to a differentiation pathway, its proliferative potential and the lifetime of the differentiated cell progeny are both strictly limited, and characterised by the loss rate Γ. Although this model admits all three possible fate outcomes, if we choose $r = 0$, we recover a process of invariant asymmetrical cell division in which each and every progenitor cell persists long-term.

Models of this kind have been studied extensively in the literature, and notably in relation to the problem of interfollicular epidermal homeostasis, where lineage tracing studies in mice show that tissue is maintained through this pattern of stochastic cell fate with $r \approx 0.1$ (*Clayton et al., 2007*). In this incarnation of the model, the fate choice is considered as a cell-autonomous or intrinsic process. Of course, in the present case, we can not rule out the possibility that the balance between proliferation and differentiation is regulated through extrinsic cues as would occur, for example, through neutral competition for limited niche space. Crucially, in the two-dimensional setting relevant to an epithelial tissue, long-term clonal evolution does discriminate between these possibilities (*Klein and Simons, 2011*). We will therefore follow this scheme, noting that the clonal fate data cannot shed light on this important issue.

Although the clonal size dependence of the model cannot be determined straightforwardly by analytic computation, we can gain intuition and recover a good approximation for the clonal dynamics by studying clonal evolution of the proliferative cell population alone. For these cells, asymmetrical cell division does not change progenitor cell number. We are therefore led to the following 'birth-death' process,

$$S \xrightarrow{2r\lambda} \begin{cases} S+S & \text{Pr} . 1/2 \\ \varnothing & \text{Pr} . 1/2' \end{cases} \tag{3}$$

For such a critical process, it is straightforward to show that the chance of finding a surviving clone with $n_S > 0$ progenitor cells at a time $t$ after the induction of single cells is given by,

$$P_{n_S}^{surv.}(t) = \frac{1}{r\lambda t}\left(1+\frac{1}{r\lambda t}\right)^{-n_S}, \tag{4}$$

with the average size of these surviving clones growing linearly with time as $n_S^{surv.}(t) \equiv \langle n_S \rangle_{surv.} = 1+r\lambda t$. At the same time, the survival probability falls $P^{surv.}(t) = 1/(1+r\lambda t)$ leading to the expected conservation law $n_s^{surv.}(t)P^{surv.}(t) = 1$, that is over time, the number of surviving clones forever diminishes while their size increases so that the average number of labelled cells remains constant.

Although these results are formally exact, they can be further simplified in the long-time limit. In particular, for $r\lambda t \gg 1$,

$$P_{n_S}^{surv.}(t) \approx \frac{1}{n_S^{surv.}(t)} \exp\left[-\frac{n_S}{n_S^{surv.}(t)}\right], \tag{5}$$

the clone size distribution acquires an exponential from. In this 'scaling limit', we can also determine a good approximation for the dynamics of the total cell population. Since each progenitor cell will be associated with a 'clonal unit' of differentiating progeny of size $1/\rho$, where $\rho$ is set by the balance between cell production and loss, $\rho\lambda = (1-\rho)\Gamma$, we have the same exponential size dependence for the total clone size,

$$P_n^{surv.}(t) \approx \frac{1}{N(t)} \exp\left[-\frac{n_S}{N(t)}\right], \tag{6}$$

where now the average size of the surviving clones grows as

$$N(t) = \frac{1+r\lambda t}{\rho}. \tag{7}$$

## On-going induction

With this platform we now turn to consider the process of on-going clone induction following the sporadic accumulation of somatic mutations. If we assume that these mutations accumulate at a constant rate,

starting at some initial time (which we define as 'time zero'), the chance that a clone will have a total of $n>0$ cells is given by (**Klein et al., 2010**)

$$P_n^{surv.}(t) = \frac{1}{\ln[\rho N(t)]} \frac{\exp[-n/N(t)]}{n}. \qquad (8)$$

while the clone survival probability is given by,

$$P^{surv.}(t) = \frac{\rho}{r\lambda t} \ln[N(t)]. \qquad (9)$$

From this last result, it follows that the clone density grows slowly (logarithmically) as

$$\sigma = RtP^{surv.}(t) = \frac{\rho R}{r\lambda} \ln[N(t)]. \qquad (10)$$

where $R$ denotes the mutation rate (per unit area). In particular, the leading dependence is on the (inverse) progenitor cell loss/replacement, $2r\lambda$. For a higher rate of loss/replacement, we expect a smaller number of clones to persist.

Although **Equation 8** provides a concrete prediction for the clone size distribution, $P_n^{surv.}(t)$, its direct application to the experimental data is compromised by fluctuations due to small number statistics. Therefore, to address the experimental data, following (**Klein et al., 2010**), we will use this result to construct a related distribution function from which a reliable comparison can be made. In particular, we may note that the cumulative sum,

$$\mu_1(n,t) = \int_0^n dm\, m P_m^{surv.}(t) \approx \mu_1(\infty,t)\left(1 - \exp[-n/N(t)]\right), \qquad (11)$$

known as the first incomplete moment, is predicted to follow an exponential form. Therefore, if we plot the quantity $1 - \mu_1(n,t)/\mu_1(\infty,t)$, we expect a pure exponential decay with a decay constant $\rho/N(t)$.

## Clonal analysis

With this result, we now turn to consider the experimental data. As discussed in the main text, we have acquired clonal fate data for six patients, three smokers and three non-smokers. First, to sensibly interpret the clonal fate data, it is necessary to make sure that the data is not compromised by clonal merger due to chance induction events at near-neighbouring sites in the tissue. For the smokers, we find a clone density of around 100 patches/cm². This translates to a typical clone separation of around $a\sim1000$ microns, around two orders of magnitude larger than the typical cell diameter. Since, on this background, the chance of finding a clone with a separation $r$ is given by

$$Y(r) = \frac{2}{a}\left(\frac{r}{a}\right) \exp\left[-\left(\frac{r}{a}\right)^2\right], \qquad (12)$$

we can deduce that the chance a clone of radius $r$ overlaps with another clone is given approximately by $\int_0^r dr\, P(r) = 1 - \exp[-(r/a)^2]$. For clones of size $n$ cells, this equates to some $\exp[-n/\kappa a^2]$, where $\kappa$ is the cell density (number per unit area). For clones of approximately 50 cells or more, this equates to around 1 in 20 clones. For these sizes, merger can play a role. However, the effect of merger is mitigated by the following effect: since small clones are more abundant, their contribution to larger clones will not significantly effect the size. But their absence from the cohort of smaller clones will influence the distribution.

With these preliminaries, we now turn to the raw clone fate data (**Figure 4—figure supplement 1A,B**). Although the first incomplete moment shows a characteristic exponential dependence over a wide range of clone sizes (**Figure 6—figure supplement 1A,B**), for the largest clone sizes, the departure of the data from exponential behavior is significant. How then, can we understand this departure, and how can we use the data to extract $N(t)$ and with it the growth parameters?

The departure at large clone sizes may derive from at least three independent sources. The first involves rare events: Although theory may correctly predict the existence of very large clones, their frequency may be very small. As a result, one would have to make a vast number of measurements to

resolve the tail of the distribution. In a typical experiment, where the number of clones is limited, such clones may not appear at all. Such rare event phenomena will give rise to strong statistical fluctuations that will plague the data for large enough clone sizes. Second, the epithelial lining of the human airways is not flat but involves a ductal structure. While clones are small, the geometry will be flat and the clone sizes will be expected to conform to the basic paradigm. However, when clone sizes become sufficiently large, the warping of the clone can also lead to constraints that confine the growth of the clone. Again, this will lead to a reduction in clone fraction at larger sizes. Thirdly, as we have seen, the infrequent chance clone merger can influence the statistics leading to a mis-representation of clone size that, by the nature of the first incomplete moment, would most adversely effect large clones. Finally, the departure of theory and data for very large clone sizes could signal a breakdown of the basic modelling scheme. For example, if tissue plays host to a rare slow-cycling stem cell population, the growth dynamics could vary. In practice, since such clones are expected to persist long term, it is likely that such a population would compromise the smaller clone sizes where exponential distribution is most robust.

Therefore, in the following, we will assume that the departure of the measured distribution from exponential is a pathology associated with rare event phenomena and small number statistics. Fortunately, to extract $N(t)$ for the majority smaller clones, we can circumvent these difficulties by focusing on a statistic closely related to the first incomplete moment. In particular, if we consider the function,

$$\frac{\mu_1(n,t)}{\mu_1(M,t)} \approx \frac{1-\exp[-n/N(t)]}{1-\exp[-M/N(t)]}, \qquad (13)$$

where $M$ denotes a large clone size cut-off beyond which we can expect rare fluctuations to dominate, we expect (and indeed observe) $\mu_1(n,t)/\mu_1(M,t)$ to depend only weakly on $M$ for a range of $M$ values. From this fit, we can obtain $N(t)$. The results are shown in *Figure 6A,B*.

## Acknowledgements

The authors appreciate the excellent technical support by Equipment Park, Experimental Pathology and Histopathology Laboratory at the London Research Institute, CRUK London, and are grateful to Professor Richard Poulsom and Dr Sagrario Cañadillas for the concept and demonstration of the practically of using porin as mitochondrial control. We thank Steve Bottoms for tissue processing and embedding, and members of the UCL Centre for Respiratory Research for helpful comments and critical evaluation of the manuscript. We are grateful to George Chennel from the scientific support services of the Wolfson Institute for Biomedical Research (UCL) and Experimental Cancer Medicine Centre. We thank Katrina McNulty for help with the manuscript preparation.

## Additional information

### Funding

| Funder | Grant reference number | Author |
|---|---|---|
| Wellcome Trust | 098357/Z/12/Z | Benjamin D Simons |
| Rosetrees Trust | | Sam M Janes |
| European Research Council | | Adam Giangreco |
| Department of Health–NIHR Biomedical Research Centre | | Adam Giangreco, Sam M Janes |
| Wellcome Trust | WT091730AIA | Sam M Janes |
| Roy Castle Lung Cancer Foundation | | Sam M Janes |

The funders had no role in study design, data collection and interpretation, or the decision to submit the work for publication.

### Author contributions

VHT, Conception and design, Acquisition of data, Analysis and interpretation of data, Drafting or revising the article; PN, Conception and design, Acquisition of data, Drafting or revising the article;

TAG, AG, BDS, SMJ, Conception and design, Analysis and interpretation of data, Drafting or revising the article; CPP, JMB, RP, DL, Drafting or revising the article, Contributed unpublished essential data or reagents; MF, EN, Acquisition of data, Drafting or revising the article; NAW, SM, Conception and design, Drafting or revising the article

**Ethics**

Human subjects: Ethical approval was sought and obtained from University College London Hospital Research Ethics Committee (REC reference 06/Q0505/12). This study was carried out in accordance with the declaration of Helsinki (2000) of the World Medical Association.

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
