## [Decision Letter]

Thank you for sending your work entitled “Stochastic homeostasis in human airway epithelium is achieved by neutral competition of basal cell progenitors” for consideration at *eLife*. Your article has been favorably evaluated by a Senior editor and 3 reviewers, one of whom is a member of our Board of Reviewing Editors, and one of whom, Patrick Warren, has agreed to reveal his identity.

The Reviewing editor and the other reviewers discussed their comments before we reached this decision, and the Reviewing editor has assembled the following comments to help you prepare a revised submission.

The reviewers agreed that your manuscript provides a novel methodology for studying lineage relationships in human tissue samples, by making use of the accumulation of somatic mutations in the mitochondrial gene, CCO. Combined with the mathematical model of clonal development and turnover, the paper provides new insights into stem and progenitor relationships in the human lung and indicates that smoking and other external influences can have an impact on clonal development.

The main major concern shared by all reviewers was the lack of clarity on the exact nature of the airway epithelial samples studied in the different analyses. We understand the constraints of obtaining human samples, but some more details on whether all samples were from exactly the same location in the lungs of different patients would be helpful. The first histological analysis appears to be on samples from bronchi and the later whole mount analysis is on “lung lobes”. How many different areas of each lung were sampled? What is the evidence that the behavior of clones in these different areas is the same? It is not inconceivable that strategies for renewal vary along the proximal-distal axis and at least this should be considered. Samples appear also to have been obtained from cancer-bearing lungs. Is there a concern that the adjacent normal epithelium may respond differently in such cases? Samples were also obtained from different ages – is there an effect of age on the clonal behavior? All of these questions can be addressed by a more detailed explanation of the clinical sampling methodology and a more careful consideration of the caveats involved in this kind of analysis.

---

## [Author Response]

*The main major concern shared by all reviewers was the lack of clarity on the exact nature of the airway epithelial samples studied in the different analyses. We understand the constraints of obtaining human samples, but some more details on whether all samples were from exactly the same location in the lungs of different patients would be helpful. The first histological analysis appears to be on samples from bronchi and the later whole mount analysis is on “lung lobes”. How many different areas of each lung were sampled? What is the evidence that the behavior of clones in these different areas is the same? It is not inconceivable that strategies for renewal vary along the proximal-distal axis and at least this should be considered. Samples appear also to have been obtained from cancer-bearing lungs. Is there a concern that the adjacent normal epithelium may respond differently in such cases? Samples were also obtained from different ages – is there an effect of age on the clonal behavior? All of these questions can be addressed by a more detailed explanation of the clinical sampling methodology and a more careful consideration of the caveats involved in this kind of analysis*.

These are important details that we agree should be clearer in the paper. The biophysical modelling analysis was based on whole mount samples obtained from the third generation bronchi. Lobectomy specimens were used that had two clear features. First the surgical cut was across the second generation bronchus (meaning the upper on the right and left; or lower lobe on the left; or lower or middle lobe on the right). The resection margin was retained by pathology and the next/third generation bronchi dissected out by the research team. Second, any resected tumour had to be in the distal lung parenchyma to avoid close proximity to the whole mount area. Importantly, the distal end of the samples were blocked and shown to be normal by H&E histology assessment. We analysed samples from seven patients. To make these points clear, we have summarised these points in the main text and referred to a more detailed discussion in a revised Materials and methods section.

To demonstrate the multipotency of the CCO deficient patches we analysed formalin fixed paraffin embedded tissue of both upper and distal airways. Although we consider the potency of cells in both upper and distal airways, the quantitative analysis of the clonal size distributions were limited to the upper airway. Therefore, to avoid the potential for confusion, we have revised Figure 1 to include only the upper airway (third generation bronchi), where we do not see Clara cells, and have transferred the discussion of multipotency in the lower airways (containing Clara cells) to a new supplementary figure. After this point the lower airways are no longer referred to in the manuscript.

On the subject of ageing, there are a number of considerations. First, the proliferative activity of stem cells and/or their differentiating progenitor cell progeny could change as a result of ageing. Second, stem cells may also change in their proliferative (self-renewal) potential: stem cells may become depleted, or other stem cells may compensate for an age-related aberrant loss of other stem cells.

To analyze the data, we have taken an objective approach and asked whether the data itself is consistent with the steady turnover of a single equipotent cell population. The observation that the clone size data conforms to an exponential size distribution characterised by a single scale, N(t), the average size of the oldest clones, suggests that the airway progenitor cell’s function as a single equipotent population over the lifetime, i.e. that the progenitor cell population does not become heterogeneous in its self-renewal potential as would be evidenced, for example, by the development of biomodality in the clone size distribution.

Secondly, by comparing the inferred value of N(t) from the data with the age of the individual patients (Figure 4), we find the average clone size of the earliest induced clones increases approximately linearly with time, suggesting that the progenitor cell loss/replacement rate remains approximately constant over time and between patients, albeit with an increased rate within smokers.

We therefore find no evidence that ageing impacts significantly on either the proliferative activity or the fate dynamics of airway progenitor cells in the period covered by this study. However, we note that an adjustment of either of these characteristics (loss/replacement rate or cell fate behaviour) in the oldest patients would not impact significantly on the clone size data (since the majority of the size dependence depends on the past history of the clone) and could therefore escape our detection.